# Whistler waves produced by monochromatic currents in the low nighttime ionosphere

Vera G. Mizonova[1,2] and Peter A. Bespalov[3]

[1]Nyzhny Novgorod State Technical University, Nyzhny Novgorod, Russia
[2]National Research University Higher School of Economics, Russia
[3]Institute of Applied Physics RAS, Nizhny Novgorod, Russia

**Correspondence:** Peter A. Bespalov
(PBespalov@mail.ru)

**Abstract.** We use a full-wave approach to find the field of monochromatic whistler waves, which are excited and propagating in the low nighttime ionosphere. The source current is located in the horizontal plane and can have arbitrary finite distribution over horizontal coordinates. The ground-based horizontal magnetic field and electric field at $125$ km are calculated. The character of wave polarization on the ground surface is investigated. The proportion in which source energy supplies the Earth-ionosphere waveguide or flows upward can be adjusted by distribution of source current. Received results are important for the analysis of ELF/VLF emission phenomena observed both on the satellites and on the ground.

## 1 Introduction

ELF/VLF waves, which propagate in the ionosphere in whistler mode, are an important part of the ionosphere dynamics. Such waves can be emitted by various natural phenomena such as atmospheric lightning discharges and volcanic eruptions, magnetospheric chorus and hiss. Artificial ELF/VLF waves have been produced by ground based transmitters and by modulated HF heating of the ionosphere current system responsible for $S_q$ variations or auroral electrojet, which is by now well-known technique. ELF/VLF waves modulated by the HAARP heating facility can be injected in the Earth-ionosphere wave-guide as far as 4400 km (Moore et al., 2007) and also into space (Inan et al., 2004).

Several numerical methods have been developed for calculating of whistler wave fields in the Earth's ionosphere (Pitteway, 1965; Wait, 1970; Bossy, 1979; Nygre'n, 1982; Budden, 1985; Nagano et al., 1994 , Yagitani et al., 1994; Shalashov and Gospodchikov, 2011). One of the main difficulties is numerical instabilities caused by a large imaginary part of the vertical wave number. General full-wave analysis, including the problem of numerical 'swamping' of the evanescent wave solutions, was made, for example, by Nygre'n (1982), Nagano et al. (1994), Budden (1985). A traditional approach in full-wave analysis is dividing a stratified ionosphere into a number of thin horizontal and homogeneous slabs and then connecting the solutions in each slab by applying the boundary conditions. Such technique has been used by Yagitani et al. (1994) to study ELF/VLF propagation from an infinitesimal dipole source located in the lower ionosphere. The idea of recursive calculation of mode amplitudes was developed and used for an arbitrary configuration of the radiating sources by Lehtinen and Inan (2008). Nevertheless, finding fields created by both natural and artificial ELF/VLF radiating sources is still very relevant.

In this paper, we use numerical methods to find the field of ELF/VLF wave, which have produced in low nighttime iono-sphere. On the one hand, significant inhomogeneity of plasma parameters, strong wave mode attenuation and effect of wave mode transformation (for example, whistler to vacuum electromagnetic) in low altitude nighttime ionosphere make the problem considered to be enough difficult and fundamentally important. On the other hand, it has practical significance, as an example, for interpretation of numerous experimental results on HF-heating which modulate natural ionospheric currents at altitudes of $60 - 100$ km.

In calculations, we use a technique known as the two-point boundary-value problem for ordinary differential equations (Kierzenka and Shampine, 2001). Using this technique in early work (Bespalov and Mizonova, 2017; Bespalov et al., 2018) has provided numerically stable solutions of a complete system of wave equations for arbitrary altitude profiles of plasma parameters and in stratified ionosphere for arbitrary angles of wave incidence from above. Here, we find a wave field created by a monochromatic source current located in the low night ionosphere. We examine the influence of peculiarities of current distribution on the proportion in which source energy supplies to the Earth-ionosphere waveguide or flows upward. As an example of calculations we use current distributions similar to those simulated by HF heating of the auroral electrojet (Payne et al., 2007). The obtained results are important for analysis of the ELF/VLF emission phenomena observed both in the ground-based observatories and on board of satellites.

## 2   Basic equations

We consider a whistler wave which is excited and propagating in the layer $0 \leq z \leq z_{\max}$ of the non-homogeneous stratified ionosphere. We choose a coordinate system with vertical upward $z$ axis and $x$, $y$ axes in horizontal plane, suppose that plasma parameters depend on coordinate $z$, plane $z = 0$ corresponds to the ground surface, above the boundary $z = z_{\max}$ ionosphere plasma is close to homogeneous, the ambient magnetic field $\mathbf{B}_0$ belongs to the $y$, $z$ plane and is inclined at an angle $\vartheta$ to the $z$ axis. We assume that external currents have monochromatic dependence on time and flow in the source plane $z = z_s$

$$\mathbf{j}(\mathbf{r}_\perp, z, t) = \mathbf{J}(\mathbf{r}_\perp)\,\delta(z - z_s)\,e^{-i\omega t}\,. \tag{1}$$

At first, we use the Fourier composition of the source current density over the horizontal coordinates

$$\mathbf{J}(\mathbf{n}_\perp, z) = \int \mathbf{J}(\mathbf{r}_\perp, z)\,e^{-ik_0 \mathbf{n}_\perp \mathbf{r}_\perp}\,k_0^2 d\mathbf{r}_\perp\,, \tag{2}$$

and wave electric and magnetic fields at each altitude

$$\mathbf{E}(\mathbf{n}_\perp, z) = \int \mathbf{E}(\mathbf{r}_\perp, z)\,e^{-ik_0 \mathbf{n}_\perp \mathbf{r}_\perp}\,k_0^2 d\mathbf{r}_\perp\,,$$

$$\mathbf{H}(\mathbf{n}_\perp, z) = \int \mathbf{H}(\mathbf{r}_\perp, z)\,e^{-ik_0 \mathbf{n}_\perp \mathbf{r}_\perp}\,k_0^2 d\mathbf{r}_\perp\,, \tag{3}$$

and find field amplitudes $\mathbf{E}(\mathbf{n}_\perp, z)$, $\mathbf{H}(\mathbf{n}_\perp, z)$ corresponding to the horizontal wave vector component $\mathbf{k}_\perp = k_0 \mathbf{n}_\perp$, $k_0 = \omega/c$. Here we use SI units for $\mathbf{E}$ and modified units for $\mathbf{H} = Z_0 \mathbf{H}_{SI}$ (Budden, 1985), where $Z_0 = \sqrt{\mu_0/\varepsilon_0}$ is the impedance of free space. Then we write the Maxwell equations

$$\nabla \times \mathbf{H} = Z_0 \mathbf{j} - ik_0 \hat{\varepsilon} \mathbf{E} \, ,$$

$$\nabla \times \mathbf{E} = ik_0 \mathbf{H} \, , \tag{4}$$

where $c$ is the speed of light in free space and $\hat{\varepsilon}$ is permittivity tensor, which yield a set of four equations for the horizontal components $\mathbf{E}_\perp(\mathbf{n}_\perp, z)$, $\mathbf{H}_\perp(\mathbf{n}_\perp, z)$ (in a case of source-free medium see, e.g., Budden, 1985; Bespalov et al., 2018; Mizonova, 2019)

$$d\mathbf{F}/dz = \hat{\mathbf{M}}\mathbf{F} + Z_0 \mathbf{I}\delta(z - z_s) \, . \tag{5}$$

Here we have taken into attention that the horizontal refractive index of the wave propagating through the stratified medium is conserved due to Snell's law. In Eq. (5) $\mathbf{F}(\mathbf{n}_\perp, z)$ and $\mathbf{I}(\mathbf{n}_\perp, z)$ are four-component column vectors

$$\mathbf{F} = \begin{pmatrix} E_x \\ E_y \\ H_x \\ H_y \end{pmatrix}, \ \mathbf{I} = \begin{pmatrix} n_x J_z/\varepsilon_{zz} \\ n_y J_z/\varepsilon_{zz} \\ J_y - J_z(\eta - \varepsilon)\sin 2\vartheta/2\varepsilon_{zz} \\ -J_x + J_z ig\sin\vartheta/\varepsilon_{zz} \end{pmatrix}, \tag{6}$$

$\hat{\mathbf{M}}$ is a matrix of which the elements $m_{ij}$ are expressed in terms of components of the transverse wave vector $\mathbf{k}_\perp = k_0 \mathbf{n}_\perp$, $\varepsilon$, $\eta$, $g$ are elements of the permittivity tensor which depends on the $z$ coordinate (Bespalov and Mizonova, 2017, Bespalov et al., 2018), $\varepsilon_{zz} = \varepsilon\sin^2\vartheta + \eta\cos^2\vartheta$.

To solve the system (5), (6) we define four boundary conditions. We write two of them on the plane $z = 0$ assuming the ground surface to be perfect conductive

$$E_x(z=0) = 0, \ \ E_y(z=0) = 0. \tag{7}$$

We write two other conditions on the plane $z = z_{\max}$ excluding wave energy coming from above. To clarify them we express the field vector column $F$ above the boundary $z = z_{\max}$ as sum of four wave modes

$$\mathbf{F}(z) = \sum_{j=1}^{4} A_j \mathbf{P}_j exp\left(ik_{zj}(z - z_{\max})\right) \, . \tag{8}$$

Here $A_j = \text{const}$, $k_{zj}$ are four roots of local dispersion relation and $\mathbf{P}_j$ are four corresponding local polarization vectors. We mention that values $k_{zj}$ and vectors $\mathbf{P}_j$ are the solution of Eqs. (5), (8) for homogeneous plasma without sources. Assuming that indices 2 and 4 correspond to coming from above propagating and non-propagating wave modes (imaginary parts of $k_{z2}$ and $k_{z4}$ are negative) we write

$$A_2 = 0, \ A_4 = 0. \tag{9}$$

Solving the set of Eq. (5) with known source current density (6) and boundary conditions (7), (9) we can find the field of plane wave with horizontal wave vector $\mathbf{k}_\perp = k_0 \mathbf{n}_\perp$ in the layer $0 < z < z_{\max}$. Then, we use the inverse transform

$$\mathbf{E}(\mathbf{r}_\perp, z) = \int \mathbf{E}(\mathbf{n}_\perp, z) e^{ik_0 \mathbf{n}_\perp \mathbf{r}_\perp} \frac{d\mathbf{n}_\perp}{(2\pi)^2} \ ,$$

$$\mathbf{H}(\mathbf{r}_\perp, z) = \int \mathbf{H}(\mathbf{n}_\perp, z) e^{ik_0 \mathbf{n}_\perp \mathbf{r}_\perp} \frac{d\mathbf{n}_\perp}{(2\pi)^2} \ , \tag{10}$$

to calculate the total field.

## 3   Description of the solution algorithm

We take into account that out of the plane $z = z_s$ the source current density (6) is equal to zero, so Eq. (5) becomes

$$d\mathbf{F}/dz = \hat{\mathbf{M}}\mathbf{F} \ . \tag{11}$$

To solve Eq. (11) in layers $0 \leq z < z_s$ and $z_s < z \leq z_{\max}$ we apply packaged solver MATLAB bvp4c and use a method of known as the two-point boundary-value problem for ordinary differential equations (Kierzenka and Shampine, 2001). The solver solution starts with an initial guess supplied at an initial mesh points and changes step-size to get the specified accuracy.

At first we find two linearly independent solutions $\mathbf{F}_1$ and $\mathbf{F}_2$ of Eq. (11) in the layer $0 \leq z < z_s$ completing the bound-

ary condition (7) on the plane $z = 0$ for arbitrary conditions on the plane $z = z_s - 0$. For example, we use four conditions $E_x(z = 0) = 0, E_y(z = 0) = 0, E_x(z = z_s - 0) = \mathfrak{E}, E_y(z = z_s - 0) = 0$ for the solution $\mathbf{F}_1$ and four conditions $E_x(z = 0) = 0, E_y(z = 0) = 0, E_x(z = z_s - 0) = 0, E_y(z = z_s - 0) = \mathfrak{E}$ for the solution $\mathbf{F}_2$, where $\mathfrak{E}$ is constant. Then we write the general solution in the layer $0 \leq z \leq z_s$ as sum

$$\mathbf{F} = a\mathbf{F}_1 + b\mathbf{F}_2 \ . \tag{12}$$

Similarly, we find two linearly independent solutions $\mathbf{F}_1^*$ and $\mathbf{F}_2^*$ of Eq. (11) in the upper layer $z_s < z \leq z_{\max}$ completing

the boundary condition (9) on the plane $z = z_{\max}$. By arbitrary conditions on the plane $z = z_s + 0$ we have $A_2 = 0$ , $A_4 = 0$, $E_x(z = z_s + 0) = \mathfrak{E}$, $E_y(z = z_s + 0) = 0$ and $A_2 = 0, A_4 = 0, E_x(z = z_s + 0) = 0, E_y(z = z_s + 0) = \mathfrak{E}$ respectively. We write the general solution in the layer $z_s < z \leq z_{\max}$ as sum

$$\mathbf{F} = a^*\mathbf{F}_1^* + b^*\mathbf{F}_2^* \ . \tag{13}$$

Integrating Eq. (5) over $z$ coordinate in a narrow layer $(z_s - 0, z_s + 0)$ we find a condition connecting solutions (12) and (13)

$$\mathbf{F}(z = z_s + 0) - \mathbf{F}(z = z_s - 0) = Z_0 \mathbf{I} \ . \tag{14}$$

That condition yields four algebraic equations for the coefficients $a$, $a^*$, $b$, $b^*$. Thus, finding those coefficients we obtain the wave field $F$ (6) in the layer $0 \leq z \leq z_{\max}$. In particular, the fields on the ground surface can be expressed as

$$H_\perp (z = 0) = \sqrt{|H_x (z = 0)|^2 + |H_y (z = 0)|^2},$$

$$E_{y'} (z = 0) = \left( \sin \vartheta - \frac{n_y}{n_{0z}} \cos \vartheta \right) n_x H_y (z = 0) -$$

$$- \left( \left( 1 - n_y^2 \right) \frac{\cos \vartheta}{n_{0z}} + n_y \sin \vartheta \right) H_x (z = 0) ,$$

$$E_{x'} (z = 0) = \frac{1 - n_x^2}{n_{0z}} H_y (z = 0) + \frac{n_x n_y}{n_{0z}} H_x (z = 0), \tag{15}$$

where $n_{0z} = \sqrt{1 - n_\perp^2}$, $E_{x',y'}$ are electric strength components of incident wave in coordinate system with $z'$ - axis along ambient magnetic field. The wave polarization near the ground surface is determined by

$$\Pi = E_{y'} (z = 0) / E_{x'} (z = 0) . \tag{16}$$

The electric field at the altitude $z = z_{\max}$ can be expressed as

$$E (z = z_{\max}) =$$

$$= \sqrt{|E_x (z = z_{\max})|^2 + |E_y (z = z_{\max})|^2 + |E_z (z = z_{\max})|^2} . \tag{17}$$

The coordinate dependence of the wave field can be found from the inverse Fourier transform (10). The vertical energy flux (Poynting vector) is

$$S_z = (2Z_0)^{-1} \mathrm{Re} \left[ \mathbf{E}_\perp^*, \mathbf{H}_\perp \right]_z , \tag{18}$$

and the total power of source is

$$P = \frac{1}{2} \mathrm{Re} \int (j_x E_x^* (z = z_s) + j_y E_y^* (z = z_s)) dx dy . \tag{19}$$

Now we present the results of numerically computed solution of the set (5), (6) under model conditions for the nighttime

ionosphere.

## 4   The ionosphere data and calculation results

In the calculations, we use the altitude profiles of the plasma density shown in Fig. 1a, and the collision frequencies between charged and neutral particles shown in Fig. 1b. The plasma density data are taken from International Reference Ionosphere (Bilitza and Reinisch, 2007) (https://ccmc.gsfc.nasa.gov/modelweb/models/iri2016_vitmo.php) and correspond to $68^0$ N; $25^0$ E;

04 September 2019, 00:30 LT. The collision frequencies data are taken from the book of Gurevich and Shvarcburg (1973). The

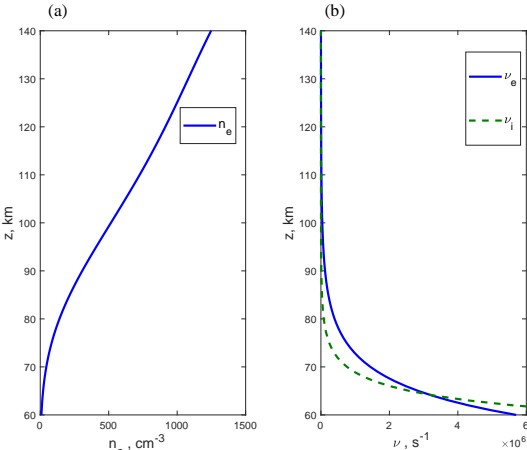

**Figure 1.** Nighttime ionosphere model: (a) the electron plasma density and (b) the collision frequency between electron and neutral particles (solid line) and collision frequency between ion and neutral particles (dashed line).

angle of magnetic field inclination with respect to the axis $z$ is equal to $\vartheta = 168^0$. We use the value $z_{\max} = 125$ km as the upper boundary of the solution. At this altitude a typical spatial scale of plasma inhomogeneity exceeds $70$ km and it is much more than the wavelength which in the considered case is of order from $20$ to $25$ km. As an example, we calculate the fields created by varying at frequency $3$ kHz and flowing at the altitude $z_s = 80$ km horizontal current. We assume that currents occupy a

5 volume which has a shape of a horizontal pancake and use for calculations a Gaussian distributions over $x$ and $y$ coordinates

$$J(\mathbf{r}_\perp) = J_{\max} \exp\left(-x^2/2L_x^2 - y^2/2L_y^2\right) . \tag{20}$$

Corresponding current distribution (2) in Fourier-space is also Gaussian and has a form $J(\mathbf{n}_\perp) = J_0 \exp\left(-k_0^2 L_x^2 n_x^2/2 - k_0^2 L_y^2 n_y^2/2\right)$, where $J_0 = 2\pi k_0^2 L_x L_y J_{\max}$. We calculate the wave field in $N = 1000$ points with steps and then use inverse Fast Fourier transform (Cooley and Tukey, 1965) to find its coordinate dependence. We present the results of field calculation in Fourier-space (Figs. 2-3) and in coordinate space (Fig. 4). The dependences of amplitude of horizontal magnetic field $H_\perp(n_x, n_y)/E_0$ on

ground surface $z = 0$ and amplitude of electric field $E(n_x, n_y)/E_0$ at altitude $z = z_{\max}$ are presented in Figs. 2a,b. The field values are normalized by the value $E_0 = Z_0 J_0$. Polarization ellipse parameters $\phi$, $E_{y'} = E_{x'} e^{-i\phi}$ and $\log|E_{y'}/E_{x'}|$ are presented in Figs. 2c,d. Positive values of phase $\phi$ correspond to right hand polarization, typical for whistler waves, and negative values of phase $\phi$ correspond to left hand polarization. Positive values of parameter $\log|E_{y'}/E_{x'}|$ mean that the polarization ellipse is elongated predominantly along $y$ axis and negative values of parameter $\log|E_{y'}/E_{x'}|$ mean that the polarization ellipse

is elongated predominantly along $x$ axis. Examples of altitude dependences of normalized electric and magnetic fields corresponding to different horizontal refractive indices are presented in Fig. 3. The level $z = z_s$ of source action is marked by dotted line. Figure 4 shows contour maps of fields created by horizontal currents (20) with characteristic horizontal sizes $L_x \simeq 12$ km,

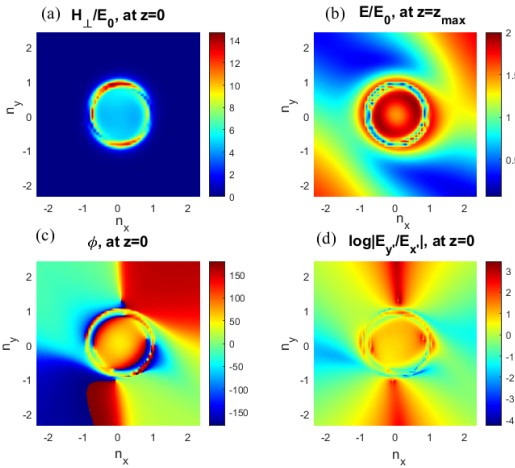

**Figure 2.** Fields in the $z = 0$ plane: (a) the horizontal magnetic field, (b) the electric field at the height $z = 125$ km, (c) the polarization ellipse parameters $\phi\left(E_{y'} = E_{x'}e^{i\phi}\right)$, and (d) $\log|E_{y'}/E_{x'}|$.

$L_y \simeq 70$ km and equal $x$ and $y$ components $J_x = J_y$ current density (Fig. 4a), electric field at the altitude $z = z_{\max}$ (Fig. 4b), horizontal magnetic field on ground surface $z = 0$ (Fig. 4c) and polarization parameter $\phi$ (Fig. 4d). Current density is normalized by the value $J_{\max}$, both electric and magnetic fields are normalized by the value $Z_0 J_{\max}$, the arrow shows the current direction. Examples of ground-based horizontal magnetic field and electric field at the altitude $z = z_{\max}$ corresponding to
the coordinate $x = 0$, characteristic horizontal sizes of source $L_y \simeq 70$ km, $L_y \simeq 12$ km and different distributions of $J_{x,y}$ are presented in Fig. 5.

## 5   Discussion

We use a full-wave approach to find the field of monochromatic whistler waves which are excited and propagating at night in the strongly inhomogeneous low ionosphere. A source current is assumed to be located in horizontal plane and to have
in this plane an arbitrary finite space distribution. At first, we consider a plane wave with the horizontal components of the refractive index $\mathbf{n}_\perp$ generated by the current $J(\mathbf{r}_\perp) \sim e^{ik_0\mathbf{n}_\perp\mathbf{r}_\perp}$. The set of wave equation in the layer $0 < z < z_{\max}$ for each $\mathbf{n}_\perp$ - component is completed by boundary conditions assuming a perfect conductivity of ground surface and excluding wave energy coming on the upper boundary $z = z_{\max}$ from above. The mathematical method known as the two-point boundary-value problem for ordinary differential equations (Kierzenka and Shampine, 2001) is applied to find the solutions of wave equations above and below the source plane. Then we connect these solutions using source current distribution. When the dependencies
of source current and wave field in $n_\perp$-space are finite functions with discretized values, the Fast Fourier Transform algorithm can be used for numerical calculations. Inverse Fast Fourier transform yields space dependence of the wave field.

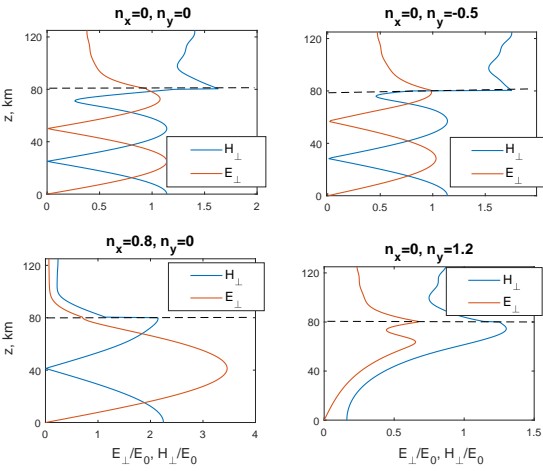

**Figure 3.** Altitude dependences of the amplitudes of horizontal electric and magnetic fields.

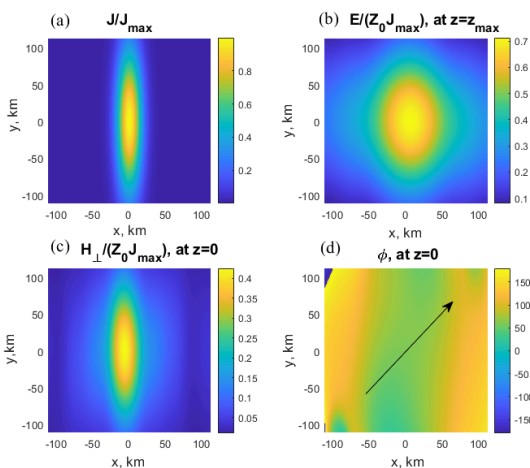

**Figure 4.** Source currents and fields space distributions: (a) space distributions of source currents, (b) the electric field at $z = 125$km, (c) the horizontal magnetic field at $z = 0$, and (d) the polarization ellipse parameters $\phi$, $E_{y'} = E_{x'}e^{-i\phi}$ at $z = 0$.

As an example, we calculate the fields created by varying at a frequency of $3$ kHz and flowing at $z_s = 80$ km horizontal current, with a Gaussian distributions of source current density over $x$ and $y$ coordinates. We mention that the model of a plane source current can also be effective in a more general case of current layer with small thickness $\Delta z \ll \lambda_z \sim 60$ km. The ground-based horizontal magnetic field and the electric field at $125$ km are calculated both in Fourier $(n_x, n_y)$ and coordinate $(x, y)$ spaces. Since a wave can achieve the ground surface in penetrating mode in case $n_\perp \leq 1$. The magnetic field $H_\perp(\mathbf{n}_\perp, z = 0)$

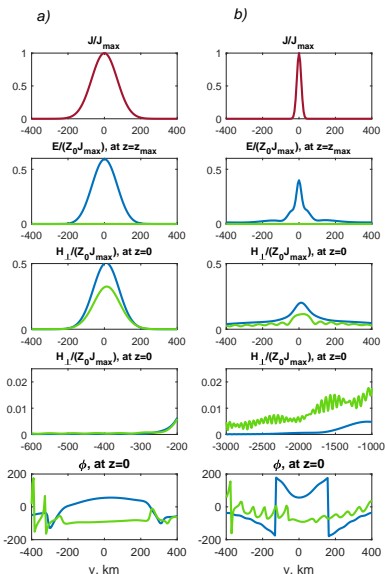

**Figure 5.** Current density (red line) and field space distributions at $x = 0$ for two different ratios $J_y/J_x$, outlined by green ($J_y/J_x = 1$) and blue ($J_y/J_x \approx -i$) colors. Left plot column (a) corresponds to $L_y = 70$ km $\geq 1/k_0$ source scale and right plot column (b) corresponds to $L_y = 12$ km $\leq 1/k_0$ source scale.

is noticeably non-zero for Fourier components with $n_\perp < 1$ and is practically equal to zero for Fourier components with $\mathbf{n}_\perp \gg 1$. Waves with $n_\perp < 1$ are right hand polarized (see Fig. 2c), which is typical for whistlers. However, if the horizontal component of the refractive index has an order of unit, then the magnetic field $H_\perp(\mathbf{n}_\perp, z = 0)$ can increase two or three times (see Fig. 2a). The polarization parameter $\phi$ of such components can be negative similar to the left hand polarized waves (see
5 Fig. 2c). A typical size of spot and a character of polarization of ground-based magnetic field $H_\perp(r_\perp, z = 0)$ depend on the source distribution. If the horizontal size of radiating currents exceeds a value $L \geq 1/k_0 \sim 15$km (for the wave of frequency 3 kHz) then the Fourier-components with $n_\perp \leq 1$ dominate in the total field. In that case, the magnetic field $H_\perp(r_\perp, z = 0)$ is predominantly localized under the source. If the horizontal size of radiating currents is small enough $L \leq 1/k_0$, then Fourier-components $n_\perp \sim 1$ can make a notable contributioninto the total field. The magnetic field $H_\perp(r_\perp, z = 0)$ occupies a spot

$\sim 1/k_0$, but can sometimes be registered far from the source (see Fig 4c, Fig. 5b). The field above the source is determined by both direct radiation from the source and the reflected radiation. Because of the effective night reflection of ELF/VLF waves, spot of the electric field at $125\,\mathrm{km}$ can exceed the size of the source by several times (Fig. 4b).

At altitudes of source and above, the two ELF/VLF wave modes (8) are weakly damped and have right hand polarization, and another two modes are evanescent and have left hand polarization. At altitudes $\sim 60 - 70\,\mathrm{km}$ and below all four modes transform into vacuum electromagnetic ones. Source currents can excite both right hand and left hand polarized modes. Hence, current distribution properties specify field distribution, field polarization and proportion in which source energy supplies the earth-ionosphere waveguide or flows upward. As an example, Figs. 5a,b show coordinate dependencies of source density, fields $E\left(x=0,y,z=0\right)$, $H_\perp\left(x=0,y,z=0\right)$ and polarization parameter $\phi$ in cases of large $L_y = 70\,\mathrm{km} \geq 1/k_0$ (a) and small $L_y = 12\,\mathrm{km} \leq 1/k_0$ (b) sources for two different ratios $J_y/J_x$ in distribution (20), outlined by green and blue colors. In the first case of ratio $J_y/J_x = 1$ ratio (blue lines on the plots), source current excite predominantly right hand polarized modes. So the polarization parameter at $z = 0$ is mainly positive under the source. Approximately 50-60 percent of the source energy is carried upward, 30-40 percent is adsorbed and about 10 percent is supplied to the Earth-ionosphere waveguide. This can be useful for modification processes in the plasma magnetic trap (Bespalov and Trakhtengerts, 1986). Of course, the electromagnetic ELF/VLF radiation of ionospheric currents themselves are not sufficient for a noticeable modification of the Earth's electron radiation belts. However, under quiet conditions in the night time magnetosphere, these emissions can ensure the radiation belts transition through the threshold of the cyclotron instability. This process can be accompanied by a significant precipitation of energetic electrons into the ionosphere and other geophysical manifestations.

In the second case of ratio $J_y/J_x \simeq -i$ ratio (green lines on the plots), source current excite predominantly left hand polarized modes. Respectively, the polarization parameter at $z = 0$ becomes predominantly negative. In that case, regardless of source size, approximately 10-12 percent of the energy is supplied to the Earth-ionosphere waveguide and about 90 percent is absorbed and not carrying upward to magnetosphere. The proportion in which source energy supplies the Earth-ionosphere waveguide or flows upward mainly depends on polarization of source current.

If the horizontal size of radiating currents is small enough $L \leq 1/k_0$, ground-based magnetic field can be noticeably non-zero far (a few thousand kilometers) from the source (see Fig. 5b). Propagating of modulated ELF/VLF signals in the Earth-ionosphere wave-guide far from the source and also into space (Inan et al., 2004) are observed experimentally. The HAARP heating facility in Gakona, Alaska injects ELF/VLF waves in the Earth-ionosphere wave-guide as far as $4400\,\mathrm{km}$ (Moore et al., 2007).

The current distributions used as example in our calculation and presented in Fig. 4 can be similar to electrojet currents modulated in D-region by the HAARP HF heating facility (Keskinen and Rowland, 2006; Payne et al., 2007). For example, according to data collected during an experimental campaign run in April 2003 and results of numerical simulations (Payne et al., 2007; Lehtinen and Inan, 2008), the maximum change in modulated conductivity occupies approximately $10\,\mathrm{km}$ over the height and occurs at altitude $\sim 80\,\mathrm{km}$. Pedersen and Hall conductivities approximately coincide so if ambient electrojet field is directed along $x$ axis, then $j_x \approx j_y$. The maximum surface density of modulated currents (1) has an order $J_\mathrm{max} \sim 10^{-6} - 10^{-5}\mathrm{Am}^{-1}$. Using in our calculations the magnitude of current density $J_\mathrm{max} \sim 5\cdot 10^{-6}\mathrm{Am}^{-1}$ yields the total power of

the source $\sim 36\,\mathrm{W}$, the ground-based horizontal magnetic field under the source $B_\perp \sim 1\mathrm{pT}$ and the electric field at the altitude of $125\,\mathrm{km}$ above the source $E \sim 400\mu\mathrm{Vm}^{-1}$. The magnitude of the magnetic field is similar to the field measured at VLF sites in the immediate vicinity of the HAARP heating facility (Payne et al., 2007) and calculated by Lehtinen and Inan (2008). The maximum vertical energy flux (Poynting vector) at the altitude of $125\,\mathrm{km}$ is $\sim 3.2\,\mathrm{nWm}^{-2}$ and total power is $\sim 17\,\mathrm{W}$. About half of the source energy is carried upward, approximately twenty percent of the energy is supplied to the Earth-ionosphere waveguide and approximately thirty percent of the energy is absorbed.

## 6 Conclusions

We find a field of monochromatic whistler waves which are excited and propagating in the low nighttime ionosphere. Using a MATLAB boundary-value problem solver enables to find numerically stable solutions of full set of the wave equations applying to conditions of inhomogeneous ionosphere at altitudes below $125\,\mathrm{km}$. Above this altitude the ionosphere plasma is slightly inhomogeneous, hence approximate methods are suitable. As example, this calculation technique is applied to the problem of ELF/VLF waves radiation from modulated HF-heated electrojet currents. At first we consider a plane wave with known horizontal component of the refractive index, find a wave field and analyze a character of wave polarization on the ground surface. Then we use inverse Fast Fourier transform to find a total field, get the dependencies of wave field at $0\,\mathrm{km}$ and $125\,\mathrm{km}$, analyze the type of wave polarization on the ground surface. The proportion in which source energy supplies the Earth-ionosphere waveguide, absorbed or flows upward depends on altitude profile of ionosphere plasma. Besides, the spatial distribution of radiating energy can be regulated by modulated currents. Depending on their properties, radiating energy can predominantly flow into space or inject into Earth-ionosphere waveguide far from the source. The obtained values are in a good agreement with ground and satellite observations and known calculation results. Using model of plane horizontal source currents can be generalized for the arbitrary altitude source distribution.

*Data availability.* The paper is theoretical and no new experimental data are used. The data are taken from International Reference Ionosphere model (Bilitza and Reinisch, 2007) (https://ccmc.gsfc.nasa.gov/modelweb/models/iri2016_vitmo.php). All figures are obtained from numerical calculation in MATLAB codes.

*Author contributions.* VM produced the calculations, analyzed results, and wrote the paper. PB proposed the problem, discussed results, and wrote the paper.

*Competing interests.* The authors declare that they have no conflict of interest.

*Acknowledgements.* The work (Sections 2-4) is supported by RFBR grant No. 20-02-00206A. The work of P.A. Bespalov (Sections 1, 5, and 6) is supported by RSF grant No. 20-12-00268. The numerical calculations were performed as part of the State Assignment of the Institute of Applied Physics RAS, project No.0035-2019-0002.

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
