# Peer review of "Whistler waves produced by monochromatic currents in the low nighttime ionosphere"

_Annales Geophysicae, 2020_

## Referee Comment (RC1) · Anonymous Referee #1 · 17 Sep 2020

Discussion of the paper by Vera G. Mizonova and Peter A. Bespalov "Whistler waves produced by monochromatic currents in the low nighttime ionosphere", submitted to ANGEO.

The paper deals with the problem of whistler mode waves excitation by a monochromatic current placed at a certain level in the lower ionosphere. The authors aim at calculating the electromagnetic field on the ground and at a certain level above the source region, using the full-wave approach. This problem is of undoubted interest for ANGEO.

I begin with minor remarks which arise when reading the manuscript.

[Figure]

1. When presenting the density profile and the profile of collision frequency, the authors refer to IRI model. As far as I know, IRI model does not provide collision frequency, at least, the URL given in the text does not lead to a site from which the collision frequency can be inferred. Thus, the given reference appears to be misleading.

2. Figure labels are too small and hardly readable.

3. Formular on page 5, line 20 is cut.

4. The sentence on page 8, line 4, starting with "Since a wave . . ." seems to be incomplete, or the dot should be replaced by a comma.

5. Page 9, line 6. Poynting vector has incorrect dimension.

6. In the Discussion and Conclusions, which follow one after another, there are almost word-for-word repetitions.

7. Although the presentation is clear, there are some mistakes in English usage. For example, p.1, line 11, "by a now" should be changed to "by now". In the Acknowledgements, "was performed" should by replaced by "were performed".

8. However, a serious point consists in the following. The authors claim that they solve the problem under conditions that "The source current is located in the horizontal plane and can have arbitrary distribution over horizontal coordinates". This claim is then repeated in Discussion. However, the authors only explain in detail how they find the field harmonics, stating that "The coordinate dependence of the wave field can be found from the inverse Fourier transform (10)" – is it that easy? It is impossible to find numerically the Fourier transform for all $n\_\perp$, thus formular (10), to which the authors refer, being suitable for analytical calculations, does not make much sense in the case of numerical ones. This point, which is commented in the text by one sentence, even few times ("Inverse Fourier transform yields space dependence of the wave field") needs a detailed explanation. How, using (10), have the authors found the field in coordinate presentation from inevitably discrete and finite number of Fourier

transforms?

---

## Author Comment (AC2) · 12 Oct 2020

The paper entitled "Whistler waves produced by monochromatic currents in the low nighttime ionosphere" [No.: angeo-2020-51] discusses an important problem concerned with a full-wave approach to find the field of monochromatic whistler waves which are excited and propagating in the low nighttime ionosphere. The source current is located in the horizontal plane and can have arbitrary distribution over horizontal coordinates. The ground-based horizontal magnetic field and electric field at large altitudes are calculated. The character of wave polarization on the ground surface is investigated. The percentages of source energy supplied by the Earth-ionosphere

waveguide and carried upward ionosphere are estimated. Received results are important for the analysis of ELF/VLF emission phenomena observed both on the satellites and on the ground.

We prepared a revised version of our paper. The reply contains author's response, marked and clean versions of the paper. The issues raised by referee have been fully addressed in the revised version and in the response, and we thus hope that the paper will now be considered acceptable for publication.

With respect, on behalf of all co-authors Peter A Bespalov IAP, Nizhny Novgorod, Russia 12 October 2020

Please also note the supplement to this comment:
https://angeo.copernicus.org/preprints/angeo-2020-51/angeo-2020-51-AC2-supplement.pdf

**Supplement:**

**(1) Comments from referee**

*The paper deals with the problem of whistler mode waves excitation by a monochromatic current placed at a certain level in the lower ionosphere. The authors aim at calculating the electromagnetic field on the ground and at a certain level above the source region, using the full-wave approach. This problem is of undoubted interest for ANGEO.*

*I begin with minor remarks which arise when reading the manuscript.*

*1.      When presenting the density profile and the profile of collision frequency, the authors refer to IRI model. As far as I know, IRI model does not provide collision frequency, at least, the URL given in the text does not lead to a site from which the collision frequency can be inferred. Thus, the given reference appears to be misleading.*

*2.      Figure labels are too small and hardly readable.*

*3.      Formular on page 5, line 20 is cut.*

*4.      The sentence on page 8, line 4, starting with "Since a wave ..." seems to be incomplete, or the dot should be replaced by a comma.*

*5.      Page 9, line 6. Poynting vector has incorrect dimension.*

*6.      In the Discussion and Conclusions, which follow one after another, there are almost word-for-word repetitions.*

*7.      Although the presentation is clear, there are some mistakes in English usage. For example, p.1, line 11, "by a now" should be changed to "by now". In the Acknowledgements, "was performed" should by replaced by "were performed".*

*8.      However, a serious point consists in the following. The authors claim that they solve the problem under conditions that "The source current is located in the horizontal plane and can have arbitrary distribution over horizontal coordinates". This claim is then repeated in Discussion. However, the authors only explain in detail how they find the field harmonics, stating that "The coordinate dependence of the wave field can be found from the inverse Fourier transform (10)" - is it that easy? It is impossible to find numerically the Fourier transform for all n_\perp, thus formular (10), to which the authors refer, being suitable for analytical calculations, does not make much sense in the case of numerical ones. This point, which is*

*commented in the text by one sentence, even few times ("Inverse Fourier transform yields space dependence of the wave field") needs a detailed explanation. How, using (10), have the authors found the field in coordinate presentation from inevitably discrete and finite number of Fourier transforms?*

**(2) Author's response**

**We would like to thank the Reviewer for the time he/she spent reading, positive response, and commenting our manuscript. We have prepared a point-by-point answer to his/her comments below. The responses are marked in bold.**

*Reviewer's Comments:*

1. *When presenting the density profile of collision frequency, the authors refer to IRI model. As far as I know, IRI model does not provide collision frequency, at least, URL given in the text does not lead to a site from which the collision frequency can be inferred. Thus, the given reference appears to be misleading.*

**Response:**

**Some clarifications concerning the collision frequency data and corresponding reference to the book of Gurevich and Shvarcburg (1973) were added to the text of the manuscript.**

*Reviewer's Comments:*

2. *Figure labels are too small and hardly readable.*

**Response:**

**Figure labels are enlarged.**

*Reviewer's Comments:*

3. *Formular on page 5, line 20 is cut*

**Response:**

**The formular was corrected.**

4. *The sentence on page 8, line 4, starting with "Since a wave…" seems to be incomplete, or the dot should be replaced by a comma.*

**Response:**

**This misprint was corrected.**

*Reviewer's Comments:*

5. *Page 9, line 6. Poynting vector has incorrect dimension.*

**Response:**

**This misprint was corrected.**

*Reviewer's Comments:*

6. *In the Discussion and Conclusions, which follow one after another, there are almost word-for-word repetitions.*

**Response:**

**The manuscript was edited.**

*Reviewer's Comments:*

7. *Although the presentation is clear, there are some mistakes in English usage. For example, p. 1, line 11, "by a now" should be changed "by now". In the Acknowledgements, "was performed" should be replaced by "were performed".*

**Response:**

**These misprints ware corrected.**

*Reviewer's Comments:*

8. *However, a serious point consist in the following. The authors claim that they solve the problem under conditions that "The source current is located in the horizontal plane and can have arbitrary distribution over horizontal coordinates". This claim is then repeated*

*in Discussions. However, the authors only explain in detail how they find the field harmonics, starting that "The coordinate dependence of the wave field can be found from the inverse Fourier transform for all n_|perp, thus formular(10), to which the authors refer, being suitable for analytical calculations, does not make much sense in the case of numerical ones. This point, which is commented in the text by one sentence, even few times ("Inverse Fourier transform yields space dependence of the wave field") needs a detailed explanation. How, using (10), have the authors found the field in coordinate presentation from inevitably discrete and finite number of Fourier transforms*

**Response:**

**We used Matlab' FFT (Fast Fourier Transform) solver. Corresponding clarification and reference were added to the text of the manuscript.**

**(3) Author's changes in manuscript**

**The modified parts are marked in yellow and the removed parts are marked in red in the new marked version of the manuscript.**

[revised manuscript text omitted]

---

## Referee Comment (RC2) · Anonymous Referee #2 · 22 Dec 2020

1. The author should clarify the motivation for this research. As has been mentioned in the second paragraph of the Introduction of this paper, "several numerical methods have been developed for calculating of whistler wave fields in the Earth's ionosphere". Then, is there any specific reason why the author chose to adopt the technique known as the two-point boundary-value problem? For example, does this technique provide more stable numerical results compared to the previously mentioned techniques? Or is this technique more efficient than other techniques?

2. Page 5, Line 10-11: "… the collision frequencies between charged and neutral particles shown in Fig.1b. The data are taken from International Reference Ionosphere…".     How are these collision frequencies obtained from IRI model? Are they calculated with some equations or from some empirical models? The related reference should be given.

3. Some typos should be corrected, for example:

    Page 4 Line 3: "… Mathlab's bvp4c…"

    Page 9 Line 1: "… $J_{max} \sim 10^{-6} \div 10^{-5} Am^{-1}$ ";

    Page 9 Line 5: " … $\sim 3,2 nWm^{-1}$ "

4. Some wording and expression need to be reconsidered? For example:

    Page 1 Line 20: "… still very actual"

    Page 5 Line 14-15: "… scale of plasma inhomogeneity exceeds 70km and is much more than…"

    Page 8 Line 14-15: "We mention that the used in our calculation current distributions…"

    Page 9 Line 9: "…By used for calculation altitude profile of …"

---

## Author Comment (AC3) · 2 Jan 2021

The comment was uploaded in the form of a supplement:
https://angeo.copernicus.org/preprints/angeo-2020-51/angeo-2020-51-AC3-supplement.zip

---

## Author Response (AR1)

Dear Editor:

We re-submit a revised version of our article entitled "Whistler waves produced by monochromatic currents in the low nighttime ionosphere" [No.: angeo-2020-51] for publication in *ANGEO-COMMUNICATIONS*. We also submit an itemized 'Author's response' to the comments offered by Reviewer#1 and Reviewer#2. The issues raised by reviewers have been fully addressed in the revised version (highlighted and clean versions of the manuscript are presented) and in the response, and we thus hope that the paper will now be considered acceptable for publication.

With highest respect,
Peter A. Bespalov
IAP, Nizhny Novgorod, Russia
2 January 2021

**(1) Comments from Reviewer#1**

*The paper deals with the problem of whistler mode waves excitation by a monochromatic current placed at a certain level in the lower ionosphere. The authors aim at calculating the electromagnetic field on the ground and at a certain level above the source region, using the full-wave approach. This problem is of undoubted interest for ANGEO.*

*I begin with minor remarks which arise when reading the manuscript.*

*1.      When presenting the density profile and the profile of collision frequency, the authors refer to IRI model. As far as I know, IRI model does not provide collision frequency, at least, the URL given in the text does not lead to a site from which the collision frequency can be inferred. Thus, the given reference appears to be misleading.*

*2.      Figure labels are too small and hardly readable.*

*3.      Formular on page 5, line 20 is cut.*

*4.      The sentence on page 8, line 4, starting with "Since a wave ..." seems to be incomplete, or the dot should be replaced by a comma.*

*5.      Page 9, line 6. Poynting vector has incorrect dimension.*

*6.      In the Discussion and Conclusions, which follow one after another, there are almost word-for-word repetitions.*

*7.      Although the presentation is clear, there are some mistakes in English usage. For example, p.1, line 11, "by a now" should be changed to "by now". In the Acknowledgements, "was performed" should by replaced by "were performed".*

*8.      However, a serious point consists in the following. The authors claim that they solve the problem under conditions that "The source current is located in the horizontal plane and can have arbitrary distribution over horizontal coordinates". This claim is then repeated in Discussion. However, the authors only explain in detail how they find the field harmonics, stating that "The coordinate dependence of the wave field can be found from the inverse Fourier transform (10)" - is it that easy? It is impossible to find numerically the Fourier transform for all n_\perp, thus formular (10), to which the authors refer, being suitable for analytical calculations, does not make much sense in the case of numerical ones. This point, which is*

*commented in the text by one sentence, even few times ("Inverse Fourier transform yields space dependence of the wave field") needs a detailed explanation. How, using (10), have the authors found the field in coordinate presentation from inevitably discrete and finite number of Fourier transforms?*

**(2) Author's response**

**We would like to thank the Reviewer#1 for the time he/she spent reading, positive response, and commenting our manuscript. We have prepared a point-by-point answer to his/her comments below. The responses are marked in bold.**

*Reviewer's Comments:*

1. *When presenting the density profile of collision frequency, the authors refer to IRI model. As far as I know, IRI model does not provide collision frequency, at least, URL given in the text does not lead to a site from which the collision frequency can be inferred. Thus, the given reference appears to be misleading.*

**Response:**

**Some clarifications concerning the collision frequency data and corresponding reference to the book of Gurevich and Shvarcburg (1973) were added to the text of the manuscript.**

*Reviewer's Comments:*

2. *Figure labels are too small and hardly readable.*

**Response:**

**Figure labels are enlarged.**

*Reviewer's Comments:*

3. *Formular on page 5, line 20 is cut*

**Response:**

**The formular was corrected.**

4. *The sentence on page 8, line 4, starting with "Since a wave…" seems to be incomplete, or the dot should be replaced by a comma.*

**Response:**

**This misprint was corrected.**

*Reviewer's Comments:*

5. *Page 9, line 6. Poynting vector has incorrect dimension.*

**Response:**

**This misprint was corrected.**

*Reviewer's Comments:*

6. *In the Discussion and Conclusions, which follow one after another, there are almost word-for-word repetitions.*

**Response:**

**The manuscript was edited.**

*Reviewer's Comments:*

7. *Although the presentation is clear, there are some mistakes in English usage. For example, p. 1, line 11, "by a now" should be changed "by now". In the Acknowledgements, "was performed" should be replaced by "were performed".*

**Response:**

**These misprints ware corrected.**

*Reviewer's Comments:*

8. *However, a serious point consist in the following. The authors claim that they solve the problem under conditions that "The source current is located in the horizontal plane and can have arbitrary distribution over horizontal coordinates". This claim is then repeated in Discussions. However, the authors only explain in detail how they find the field harmonics, starting that "The coordinate dependence of the wave field can be found from the inverse Fourier transform for all $n\_{\perp}$, thus formular(10), to which the authors refer, being suitable for analytical calculations, does not make much sense in the case of numerical ones. This point, which is commented in the text by one sentence, even few times ("Inverse Fourier transform yields space dependence of the wave field") needs a detailed explanation. How, using (10), have the authors found the field in coordinate presentation from inevitably discrete and finite number of Fourier transforms*

**Response:**

**We used MATLAB FFT (Fast Fourier Transform) solver. Corresponding clarification and reference were added to the text of the manuscript.**

**(3) Author's changes in manuscript**

**The modified parts are marked in yellow and the removed parts are marked in red in the new marked version of the manuscript.**

[revised manuscript text omitted]

**(1) Comments from Reviewer#2**

*1. The author should clarify the motivation for this research. As has been mentioned in the second paragraph of the Introduction of this paper, "several numerical methods have been developed for calculating of whistler wave fields in the Earth's ionosphere". Then, is there any specific reason why the author chose to adopt the technique known as the two-point boundary-value problem? For example, does this technique provide more stable numerical results compared to the previously mentioned techniques? Or is this technique more efficient than other techniques?*

*2. Page 5, Line 10-11: "… the collision frequencies between charged and neutral particles shown in Fig.1b. The data are taken from International Reference Ionosphere…". How are these collision frequencies obtained from IRI model? Are they calculated with some equations or from some empirical models? The related reference should be given.*

*3. Some typos should be corrected, for example:*

*Page 4 Line 3: "… Mathlab's bvp4c…"*

*Page 9 Line 1: "… $J_{max} \sim 10^{-6} \div 10^{-5} \ Am^{-1}$ ";*

*Page 9 Line 5: " … $\sim 3,2nWm^{-1}$ "*

*4. Some wording and expression need to be reconsidered? For example:*

*Page 1 Line 20: "… still very actual"*

*Page 5 Line 14-15: "… scale of plasma inhomogeneity exceeds 70km and is much more than…"*

*Page 8 Line 14-15: "We mention that the used in our calculation current distributions…"*

*Page 9 Line 9: "…By used for calculation altitude profile of …"*

**(2) Author's response**

**We would like to thank the Reviewer#2 for the time he/she spent reading, positive response, and commenting our manuscript. We have prepared a point-by-point answer to his/her comments below. The responses are marked in bold.**

*Reviewer's Comments:*

> *1. The author should clarify the motivation for this research. As has been mentioned in the second paragraph of the Introduction of this paper, "several numerical methods have been developed for calculating of whistler wave fields in the Earth's ionosphere". Then, is there any specific reason why the author chose to adopt the technique known as the two-point boundary-value problem? For example, does this technique provide more stable numerical results compared to the previously mentioned techniques? Or is this technique more efficient than other techniques?*

**Response:**

**Some clarifications devoted to the better motivation for this research were added to the text of the Introduction.**

**The text**

**"General full-wave analysis, including the problem of numerical 'swamping' of the evanescent wave solutions, was made, for example, by Nygre'n (1982), Nagano et al. (1994), Budden (1985). Full wave calculation of ELF/VLF propagationfrom a dipole source located in the lower ionosphere has been made by Yagitani et al. (1994) The idea of recursive calculation of mode amplitudes was developed and used for an arbitrary configuration of the radiating sources by Lehtinen and Inan (2008). Nevertheless, finding fields created by both natural and artificial ELF/VLF radiating sources is still very actual.**

**In this paper, we use numerical methods to find the field of a whistler wave generated and propagating in low night ionosphere. We use a technique known as the two-point boundary-value problem for ordinary differential equations (Kierzenka and Shampine, 2001). Using this technique in early work (Bespalov and Mizonova, 2017; Bespalov et al., 2018) has provided numerically stable solutions of a complete system of wave equations for arbitrary altitude profiles of plasma parameters and for arbitrary angles of wave incidence."**

**is replaced by**

**"General full-wave analysis, including the problem of numerical 'swamping' of the evanescent wave solutions, was made, for example, by Nygre'n (1982), Nagano et al. (1994), Budden**

(1985). A traditional approach in full-wave analysis is dividing a stratified ionosphere into a number of thin horizontal and homogeneous slabs and then connecting the solutions in each slab by applying the boundary conditions. Such technique has been used by Yagitani et al. (1994) to study ELF/VLF propagation from an infinitesimal dipole source located in the lower ionosphere. The idea of recursive calculation of mode amplitudes was developed and used for an arbitrary configuration of the radiating sources by Lehtinen and Inan (2008). Nevertheless, finding fields created by both natural and artificial ELF/VLF radiating sources is still very relevant.

In this paper, we use numerical methods to find the field of ELF/VLF wave, which have parameters, strong wave mode attenuation and effect of wave mode transformation (for example, whistler to vacuum electromagnetic) in low altitude nighttime ionosphere make the problem considered to be enough difficult and fundamentally important. On the other hand, it has practical significance, as an example, for interpretation of numerous experimental results on HF-heating which modulate natural ionospheric currents at altitudes of 60-100 km.

In calculations, we use a technique known as the two-point boundary-value problem for ordinary differential equations (Kierzenka and Shampine, 2001). Using this technique in early work (Bespalov and Mizonova, 2017; Bespalov et al., 2018) has provided numerically stable solutions of a complete system of wave equations for arbitrary altitude profiles of plasma parameters and in stratified ionosphere for arbitrary angles of wave incidence."

*Reviewer's Comments:*

*2. Page 5, Line 10-11: "… the collision frequencies between charged and neutral particles shown in Fig.1b. The data are taken from International Reference Ionosphere…". How are these collision frequencies obtained from IRI model? Are they calculated with some equations or from some empirical models? The related reference should be given.*

**Response:**

**Some clarifications concerning the collision frequency data and corresponding reference to the book of Gurevich and Shvarcburg (1973) were added to the text of the manuscript in accordance with Reviewer1's Comment.**

*Reviewer's Comments:*

*3. Some typos should be corrected, for example:*

*Page 4 Line 3: "... Mathlab's bvp4c..."*

*Page 9 Line 1: "... $J_{\max} \sim 10^{-6} \div 10^{-5} \ Am^{-1}$ ";*

*Page 9 Line 5: "... $\sim 3,2 nWm^{-1}$ "*

**Response:**

**The typos "... *Mathlab's bvp4c...*" and "... $\sim 3,2 nWm^{-1}$ " were corrected. Estimation**
$J_{\max} \sim 10^{-6} \div 10^{-5} \ Am^{-1}$ **was made for surface current density, the reference to corresponding Eq. (1) was added to the text.**

*Reviewer's Comments:*

*4. Some wording and expression need to be reconsidered? For example:*

*Page 1 Line 20: "... still very actual"*

*Page 5 Line 14-15: "... scale of plasma inhomogeneity exceeds 70km and is much more than..."*

*Page 8 Line 14-15: "We mention that the used in our calculation current distributions..."*

*Page 9 Line 9: "...By used for calculation altitude profile of ..."*

**Response:**

**This wording and expression were reconsidered.**

**(3) Author's changes in manuscript**

**The modified parts are marked in green and the removed parts are marked in red in the new marked version of the manuscript.**

[revised manuscript text omitted]

---

## Author Response (AR2)

Dear Editor:

We re-submit a revised version of our article entitled "Whistler waves produced by monochromatic currents in the low nighttime ionosphere" [No.: angeo-2020-51] for publication in *ANGEO-COMMUNICATIONS*. We highlighted the progress achieved in the article compared to the results already published, performed additional calculations and added new material (see Fig. 5 and its description). Our results show the possibility of controlling the energy fluxes of electromagnetic ELF/VLV radiation entering the Earth-ionosphere waveguide and the magnetosphere. We also submit an itemized 'Author's response' to the comments offered by Reviewer after the second revision. The issues raised by reviewer have been fully addressed in the revised version (highlighted and clean versions of the manuscript are presented) and in the response, and we thus hope that the paper will now be considered acceptable for publication.

With highest respect,
Peter A. Bespalov
IAP, Nizhny Novgorod, Russia
3 May 2021

**(1) Comments from referee**

*This study is about VLF propagation calculated within the full wave model. Authors adopts already well known technique (namely "two-point boundary-value MATLAB solver" by Kierzenka and Shampine (2001)) to quite model problem of VLF generation by plane currents. Although the topic is generally interesting, I did not find anything particular new in this study... basically Authors report that the two-point boundary-value Maltab solver works well, but this is not scientific result. To shape a scientifically sounding study, Authors would need either provide some comparison with spacecraft/ground-based measurements, or demonstrate new theoretical results (and describe in details what new has been done). The present form of this study does not contain such comparisons or new theories. Thus, I cannot recommend it for publication.*

**(2) Author's response**

**We would like to thank the Reviewer for the time he/she spent reading, and commenting our manuscript. We have prepared a point-by-point answer to his/her comments below. The responses are marked in bold.**

*Reviewer's Comments:*

*This study is about VLF propagation calculated within the full wave model. Authors adopts already well known technique (namely "two-point boundary-value MATLAB solver" by Kierzenka and Shampine (2001)) to quite model problem of VLF generation by plane currents.*

**Response:**

**The article by Kierzenka and Shampine (2001) implements a purely mathematical algorithm for solving a boundary value problem. There was no geophysics there at all. In this work, we used this algorithm for the first time to solve an important geophysical problem closely related to active experiments on short-wave ionospheric heating facility. Other researchers have considered close problems by methods that are less perfect from a mathematical point of view, without guaranteeing the convergence and accuracy of the calculation results.**

*Reviewer's Comments:*

*Although the topic is generally interesting, I did not find anything particular new in this study... basically Authors report that the two-point boundary-value Maltab solver works well, but this is not scientific result. To shape a scientifically sounding study, Authors would need either provide some comparison with spacecraft/ground-based measurements, or demonstrate new theoretical*

*results (and describe in details what new has been done). The present form of this study does not contain such comparisons or new theories. Thus, I cannot recommend it for publication.*

**Response:**

**The adaptation and use of an advanced mathematical apparatus made it possible to compactly formulate an important geophysical problem, taking into account the real inhomogeneity of the parameters of the ionosphere and to advance the calculations to a new level. Our results show the possibility of controlling the energy fluxes of electromagnetic radiation entering the Earth-ionosphere waveguide and the magnetosphere. The revised version of the manuscript provides an example of the distribution of external ionospheric currents, which do not direct the energy into the magnetosphere (see green lines in Fig. 5). We have expanded the bibliography to compare results with ground-based measurements.**

**On the contrary, it is possible increase the level of energy directed into the magnetosphere (see blue lines in Fig. 5). This can be useful for modification processes in the plasma magnetic trap. Of course, the electromagnetic ELF/VLF radiation of ionospheric currents themselves are not sufficient for a noticeable modification of the Earth's electron radiation belts. However, under quiet conditions in the nighttime magnetosphere, these emissions can ensure the radiation belts transition through the threshold of the cyclotron instability. This process can be accompanied by a significant precipitation of energetic electrons into the ionosphere and other geophysical manifestations.**

**(3) Author's changes in manuscript**

**The modified parts are marked in yellow and the removed parts are marked in red in the new marked version of the manuscript.**